# FedComLoc: Communication-Efficient Distributed Training of Sparse and Quantized Models

**Kai Yi**[*]                                                                    *kai.yi@kaust.edu.sa*
*Department of Computer Science*
*King Abdullah University of Science and Technology (KAUST)*

**Georg Meinhardt**[†]                                              *georg.meinhardt@kaust.edu.sa*
*Department of Computer Science*
*King Abdullah University of Science and Technology (KAUST)*

**Laurent Condat**                                                  *laurent.condat@kaust.edu.sa*
*Department of Computer Science*
*King Abdullah University of Science and Technology (KAUST)*
*SDAIA-KAUST Center of Excellence in Data Science and Artificial Intelligence (SDAIA-KAUST AI)*

**Peter Richtárik**                                                 *peter.richtarik@kaust.edu.sa*
*Department of Computer Science*
*King Abdullah University of Science and Technology (KAUST)*
*SDAIA-KAUST Center of Excellence in Data Science and Artificial Intelligence (SDAIA-KAUST AI)*

**Reviewed on OpenReview:** *https://openreview.net/forum?id=vYQPLytQsj*

## Abstract

Federated Learning (FL) has garnered increasing attention due to its unique characteristic of allowing heterogeneous clients to process their private data locally and interact with a central server, while being respectful of privacy. A critical bottleneck in FL is the communication cost. A pivotal strategy to mitigate this burden is *Local Training*, which involves running multiple local stochastic gradient descent iterations between communication phases. Our work is inspired by the innovative Scaffnew algorithm of Mishchenko et al. (2022), which has considerably advanced the reduction of communication complexity in FL. We introduce FedComLoc (Federated Compressed and Local Training), integrating practical and effective compression into Scaffnew to further enhance communication efficiency. Extensive experiments, using the popular TopK compressor and quantization, demonstrate its prowess in substantially reducing communication overheads in heterogeneous settings.

## 1 Introduction

Privacy concerns and limited computing resources on edge devices often make centralized training impractical, where all data is gathered in a data center for processing. In response to these challenges, Federated Learning (FL) has emerged as an increasingly popular framework (McMahan et al., 2016; Kairouz et al., 2019). In FL, multiple clients perform computations locally on their private data and exchange information with a central server. This process is typically framed as an empirical risk minimization problem (Shalev-Shwartz & Ben-David, 2014):

$$\min_{x \in \mathbb{R}^d} \left[ f(x) := \frac{1}{n} \sum_{i=1}^{n} f_i(x) \right], \qquad \text{(ERM)}$$

---

[*]Work completed during PhD at KAUST. Currently with Meta, USA.
[†]Work done while an intern at KAUST.

where $f_i$ represents the local objective for client $i$, $n$ is the total number of clients, and $x$ is the model to be optimized. Our primary objective is to solve the problem (ERM) and deploy the optimized global model to all clients. For instance, $x$ might be a neural network trained in an FL setting. However, a considerable bottleneck in FL are communication cost, particularly with large models.

To mitigate these costs, FL often employs *Local Training* (LT), a strategy where local parameters are updated multiple times before aggregation (Povey et al., 2014; Moritz et al., 2016; McMahan et al., 2017; Li et al., 2020; Haddadpour & Mahdavi, 2019; Khaled et al., 2019; 2020; Karimireddy et al., 2020; Gorbunov et al., 2020; Mitra et al., 2021). However, there is a lack of theoretical understanding regarding the effectiveness of LT methods. The recent introduction of Scaffnew by Mishchenko et al. (2022) marked a substantial advancement, as this algorithm converges to the exact solution with accelerated complexity, in convex settings.

Another approach to reducing communication costs is through compression (Haddadpour et al., 2021; Condat et al., 2022; Yi et al., 2024). In centralized training, one often aims to learn a sparsified model for faster training and communication efficiency (Dettmers & Zettlemoyer, 2019; Kuznedelev et al., 2023). Dynamic pruning strategies like gradual magnitude pruning (Gale et al., 2019) and RigL (Evci et al., 2020) are common. But in FL, the effectiveness of these model sparsification methods based on thresholds remains unclear. The work by Babakniya et al. (2023) considers FL sparsity mask concepts, showing promising results.

Quantization is another efficient model compression technique (Han et al., 2021; Bhalgat et al., 2020; Shin et al., 2023), though its application in heterogeneous settings is limited. Gupta et al. (2022) introduced FedAvg with Kurtosis regularization (Chmiel et al., 2020) in FL.

Furthermore, studies such as Haddadpour et al. (2021); Condat et al. (2022) have theoretical convergence guarantees for unbiased estimators with restrictive assumptions. As this work employs the biased TopK compressor these are unsuitable in this case.

Thus, we tackle the following question:

*Is it possible to design an efficient algorithm combining accelerated local training with compression techniques, such as quantization and Top-K, and validate its efficiency empirically on popular FL datasets?*

Our method for investigating this question consists of two steps. Firstly, we design an algorithm, termed FedComLoc, which integrates general compression into ScaffNew, an accelerated local training algorithm. Secondly, we empirically validate FedComLoc for popular compression techniques (Top$K$ and quantization) on popular FL datasets (FedMNIST and FedCIFAR10).

We were able to answer this question affirmatively with the following contributions:

- We have developed a communication-efficient method FedComLoc for distributed training, specifically designed for heterogeneous environments. This method integrates general compression techniques and is motivated by previous theoretical insights.

- We proposed three variants of our algorithm addressing several key bottlenecks in FL: FedComLoc-Com addresses communication costs from client to server, FedComLoc-Global addresses communication costs from server to client and FedComLoc-Local addresses limited computational resources on edge devices.

- We conducted detailed comparisons and ablation studies, validating the effectiveness of our approach. These reveal a considerable reduction in communication and, in certain cases, an enhancement in training speed in number of communication rounds. Furthermore, we demonstrated that our method outperforms well-established baselines in terms of training speed and communication costs.

## 2 Related Work

### 2.1 Local Training

The evolution of LT in FL has been profound and continuous, transitioning through five distinct generations, each marked by considerable advancements from empirical discoveries to reductions in communication

---

**Algorithm 1** FedComLoc

---

1: stepsize $\gamma > 0$, probability $p > 0$, initial iterate $x_{1,0} = \cdots = x_{n,0} \in \mathbb{R}^d$, initial control variates $h_{1,0}, \ldots, h_{n,0} \in \mathbb{R}^d$ on each client such that $\sum_{i=1}^n h_{i,0} = 0$, number of iterations $T \geq 1$, compressor $\mathrm{C}(\cdot) \in \{\mathrm{Top}K(\cdot), \mathrm{Q}_\mathrm{r}(\cdot), \cdots\}$

2: **server:** flip a coin, $\theta_t \in \{0, 1\}$, $T$ times, where $\mathrm{Prob}(\theta_t = 1) = p$ ⋄ Decide when to skip communication

3: send the sequence $\theta_0, \ldots, \theta_{T-1}$ to all workers

4: **for** $t = 0, 1, \ldots, T - 1$ **do**

5:     sample clients $\mathcal{S} \subseteq \{1, 2, 3, \ldots, n\}$

6:     **in parallel on all workers** $i \in \mathcal{S}$ **do**

7:         FedComLoc-Local: local compression – $g_{i,t}(x_{i,t}) = g_{i,t}(\mathrm{C}(x_{i,t}))$

8:         $\hat{x}_{i,t+1} = x_{i,t} - \gamma(g_{i,t}(x_{i,t}) - h_{i,t})$    ⋄ Local gradient-type step adjusted via the local control variate $h_{i,t}$

9:         FedComLoc-Com: uplink compression – $\hat{x}_{i,t+1} = \mathrm{C}(\hat{x}_{i,t+1})$

10:         **if** $\theta_t = 1$ **then**

11:             $x_{i,t+1} = \frac{1}{n} \sum_{i=1}^n \hat{x}_{i,t+1}$       ⋄ Average the iterates (with small probability $p$)

12:             FedComLoc-Global: downlink compression – $x_{i,t+1} = \mathrm{C}(x_{i,t+1})$

13:         **else**

14:             $x_{i,t+1} = \hat{x}_{i,t+1}$              ⋄ Skip communication

15:         **end if**

16:         $h_{i,t+1} = h_{i,t} + \frac{p}{\gamma}(x_{i,t+1} - \hat{x}_{i,t+1})$       ⋄ Update the local control variate $h_{i,t}$

17:     **end local updates**

18: **end for**

---

complexity. The pioneering FedAvg algorithm (McMahan et al., 2017) represents the first generation of LT techniques, primarily focusing on empirical evidence and practical applications (Povey et al., 2014; Moritz et al., 2016; McMahan et al., 2017). The second generation of LT methods consists in solving (ERM) based on homogeneity assumptions such as bounded gradients[1] (Li et al., 2020) or limited gradient diversity[2] (Haddadpour & Mahdavi, 2019). However, the practicality of such assumptions in real-world FL scenarios is debatable and often not viable (Kairouz et al., 2019; Wang et al., 2021).

Third-generation methods made fewer assumptions, demonstrating sublinear (Khaled et al., 2019; 2020) or linear convergence up to a neighborhood (Malinovsky et al., 2020) with convex and smooth functions. More recently, fourth-generation algorithms like Scaffold (Karimireddy et al., 2020), S-Local-GD (Gorbunov et al., 2020), and FedLin (Mitra et al., 2021) have gained popularity. These algorithms effectively counteract client drift and achieve linear convergence to the exact solution under standard assumptions. Despite these advances, their communication complexity mirrors that of GD, i.e. $\mathcal{O}(\kappa \log \epsilon^{-1})$, where $\kappa := L/\mu$ denotes the condition number.

The most recent Scaffnew algorithm, proposed by Mishchenko et al. (2022), revolutionizes the field with its accelerated communication complexity $\mathcal{O}(\sqrt{\kappa} \log \epsilon^{-1})$. This seminal development establishes LT as a communication acceleration mechanism for the first time, positioning Scaffnew at the forefront of the fifth generation of LT methods with accelerated convergence. Further enhancements to Scaffnew have been introduced, incorporating aspects like variance-reduced stochastic gradients (Malinovsky et al., 2022), personalization (Yi et al., 2023), partial client participation (Condat et al., 2023), asynchronous communication (Maranjyan et al., 2022), and an expansion into a broader primal–dual framework (Condat & Richtárik, 2023). This latest generation also includes the 5GCS algorithm (Grudzień et al., 2023), with a different strategy where the local steps are part of an inner loop to approximate a proximity operator. Our proposed FedComLoc algorithm extends Scaffnew by incorporating pragmatic compression techniques, such as sparsity and quantization, resulting in even faster training measured by the number of bits communicated.

---

[1] There exists $c \in \mathbb{R}$ s.t. $\|\nabla f_i(x)\| \leq c$ for $1 \leq i \leq d$.

[2] There exists $c \in \mathbb{R}$ s.t. $\frac{1}{n} \sum_{i=1}^n \|\nabla f_i(x)\|^2 \leq c\|\nabla f(x)\|^2$.

## 2.2 Model Compression in Federated Learning

Model compression in the context of FL is a burgeoning field with diverse research avenues, particularly focusing on the balance between model efficiency and performance. Jiang et al. (2022) innovated in global pruning by engaging a single, powerful client to initiate the pruning process. This strategy transitions into a collaborative local pruning phase, where all clients contribute to an adaptive pruning mechanism. This involves not just parameter elimination, but also their reintroduction, integrated with the standard FedAvg framework (McMahan et al., 2016). However, this approach demands substantial local memory for tracking the relevance of each parameter, a constraint not always feasible in FL settings.

Addressing some of these challenges, Huang et al. (2022) introduced an adaptive batch normalization coupled with progressive pruning modules, enhancing sparse local computations. These advancements, however, often do not fully address the constraints related to computational resources and communication bandwidth on the client side. Our research primarily focuses on magnitude-based sparsity pruning. Techniques like gradual magnitude pruning (Gale et al., 2019) and RigL (Evci et al., 2020) have been instrumental in dynamic pruning strategies. However, their application in FL contexts remains relatively unexplored. The pioneering work of Babakniya et al. (2023) extends the concept of sparsity masks in FL, demonstrating noteworthy outcomes.

Quantization is another vital avenue in model compression. Seminal works in this area include Han et al. (2021), Bhalgat et al. (2020), and Shin et al. (2023). A major advance has been made by Gupta et al. (2022), who combined FedAvg with Kurtosis regularization (Chmiel et al., 2020). We are looking to go even further by integrating accelerated LT with quantization techniques.

However, a gap exists in the theoretical underpinnings of these compression methods. Research by Haddad-pour et al. (2021) and Condat et al. (2022) offers theoretical convergence guarantees for unbiased estimators, but these frameworks are not readily applicable to common compressors like Top-K sparsifiers. In particular, CompressedScaffnew (Condat et al., 2022) integrates an unbiased compression mechanism in Scaffnew, that is based on random permutations. But due to requiring shared randomness it is not practical. Linear convergence has been proved when all functions $f_i$ are strongly convex.

To the best of our knowledge, no other compression mechanism has been studied in Scaffnew, either theoretically or empirically, and even the mere convergence of Scaffnew in nonconvex settings has not been investigated either. Our goal is to go beyond the convex setting and simplistic logistic regression experiments and to study compression in Scaffnew in realistic nonconvex settings with large datasets such as Federated CIFAR and MNIST. Our integration of compression in Scaffnew is heuristic but backed by the findings and theoretical guarantees of CompressedScaffnew in the convex setting, which shows a twofold acceleration with respect to the conditioning $\kappa$ and the dimension $d$, thanks to LT and compression, respectively.

## 3 Proposed Algorithm FedComLoc

### 3.1 Sparsity and Quantization

Let us define the sparsifying $\mathrm{Top}K(\cdot)$ and quantization $\mathrm{Q_r}(\cdot)$ operators.

**Definition 1.** Let $x \in \mathbb{R}^d$ and $K \in \{1, 2, \ldots, d\}$. We define the sparsifying compressor $\mathrm{Top}K : \mathbb{R}^d \to \mathbb{R}^d$ as:

$$\mathrm{Top}K(x) \coloneqq \arg\min_{y \in \mathbb{R}^d} \{\|y - x\| \mid \|y\|_0 \leq K\},$$

where $\|y\|_0 \coloneqq |\{i : y_i \neq 0\}|$ denotes the number of nonzero elements in the vector $y = (y_1, \cdots, y_d)^\intercal \in \mathbb{R}^d$. In case of multiple minimizers, $\mathrm{Top}K$ is chosen arbitrarily.

**Definition 2.** For any vector $x \in \mathbb{R}^d$, with $x \neq \mathbf{0}$ and a number of bits $r > 0$, its binary quantization $\mathrm{Q_r}(x)$ is defined componentwise as

$$\mathrm{Q_r}(x) = \left(\|x\|_2 \cdot \mathrm{sgn}(x_i) \cdot \xi_i(x, 2^r)\right)_{1 \leq i \leq d},$$

where $\xi_i(x, 2^r)$ are independent random variables. Let $y_i := \frac{|x_i|}{\|x\|_2}$. Then their probability distribution is given by

$$\xi_i(x, 2^r) = \begin{cases} \lceil 2^r y_i \rceil / 2^r & \text{with proba. } 2^r y_i - \lfloor 2^r y_i \rfloor ; \\ \lfloor 2^r y_i \rfloor / 2^r & \text{otherwise.} \end{cases}$$

If $x = \mathbf{0}$, we define $Q_r(x) = \mathbf{0}$.

The distributions of the $\xi_i(x, r)$ minimize variance over distributions with support $\{0, 1/r, \ldots, 1\}$, ensuring unbiasedness, i.e. $\mathbb{E}[\xi_i(x, r)] = |x_i| / \|x\|_2$. This definition is based on an equivalent one in Alistarh et al. (2017).

### 3.2 Introduction of the Algorithms

FedComLoc (Algorithm 1) is an adaptation of the accelerated Scaffnew framework, with modifications to incorporate communication-efficient compression strategies. In this section, we detail how sparsification and quantization are integrated, and we explain the design and motivation behind each FedComLoc variant.

In all variants, federated training proceeds in rounds: the server broadcasts the current global model, clients perform local updates, and compressed updates are communicated as appropriate. We adopt $\text{Top}K(\cdot)$ as the default compression technique, but note that quantization can be used interchangeably.

We consider three variants, each targeting a different practical bottleneck in federated systems:

- FedComLoc-Com **(Client-to-Server Compression):** This variant addresses the communication bottleneck on the uplink. After each client completes its local training, the update or local model is compressed (via TopK or quantization) before being transmitted to the server. This approach is well-suited for scenarios where client devices are bandwidth-limited for uploading. The server then aggregates the compressed updates to form the next global model.

- FedComLoc-Local **(Local Model Compression):** In this setting, compression is applied within the local training loop. After each local update step (e.g., after a gradient step or batch), the client compresses its own local model or updates. This variant is designed for cases where client devices have limited computational or memory resources, and thus benefit from working with compressed models during training. It can also reduce local energy consumption and memory footprint.

- FedComLoc-Global **(Server-to-Client Compression):** Here, the server compresses the global model before sending it to all clients at the start of each round. This variant targets scenarios where the main bottleneck is the download bandwidth from server to clients (e.g., in environments with many clients or costly downlink channels). Each client then decompresses the received model before starting local training.

Algorithm 1 (see above) provides a unified pseudocode for all three variants. The specific compression step is invoked at the corresponding stage (client upload, local update, or server broadcast) depending on the variant in use.

These variants allow FedComLoc to flexibly adapt to different system bottlenecks commonly encountered in federated learning applications. Detailed ablation results in the following experiments demonstrate the practical impact of each approach.

## 4 Experiments

**Baselines.** Our evaluation comprises three distinct aspects. Firstly, we conduct experiments to assess the impact of compression on communication costs. FedComLoc is assessed for varying sparsity and quantization ratios. Secondly, we compare FedComLoc-Com with FedComLoc-Local and FedComLoc-Global. Thirdly, we explore the efficacy of FedComLoc against non-accelerated local training methods, including FedAvg (McMahan et al., 2016), its Top-K sparsified counterpart sparseFedAvg, and Scaffold (Karimireddy et al., 2020).

| Top-K | 100% | 10% | 30% | 50% | 70% | 90% |
|---|---|---|---|---|---|---|
| Accuracy | 0.9758 | 0.9374 | 0.9654 | 0.9699 | 0.9745 | 0.9748 |
| Decrease | - | 3.94% | 1.07% | 0.61% | 0.13% | 0.10% |

Table 1: Test accuracy for various Top-K ratios.

| | $\alpha = 0.1$ | $\alpha = 0.3$ | $\alpha = 0.5$ | $\alpha = 0.7$ | $\alpha = 0.9$ | $\alpha = 1.0$ |
|---|---|---|---|---|---|---|
| K =100% | 0.9623 | 0.9686 | 0.9731 | 0.9758 | 0.9768 | 0.9735 |
| K =10% | 0.8681 | 0.9124 | 0.9331 | 0.9374 | 0.9441 | 0.9382 |
| K =50% | 0.9597 | 0.9635 | 0.9671 | 0.9699 | 0.9706 | 0.9719 |

Table 2: Test accuracy score for various Dirichlet factors $\alpha$ and sparsity ratios.

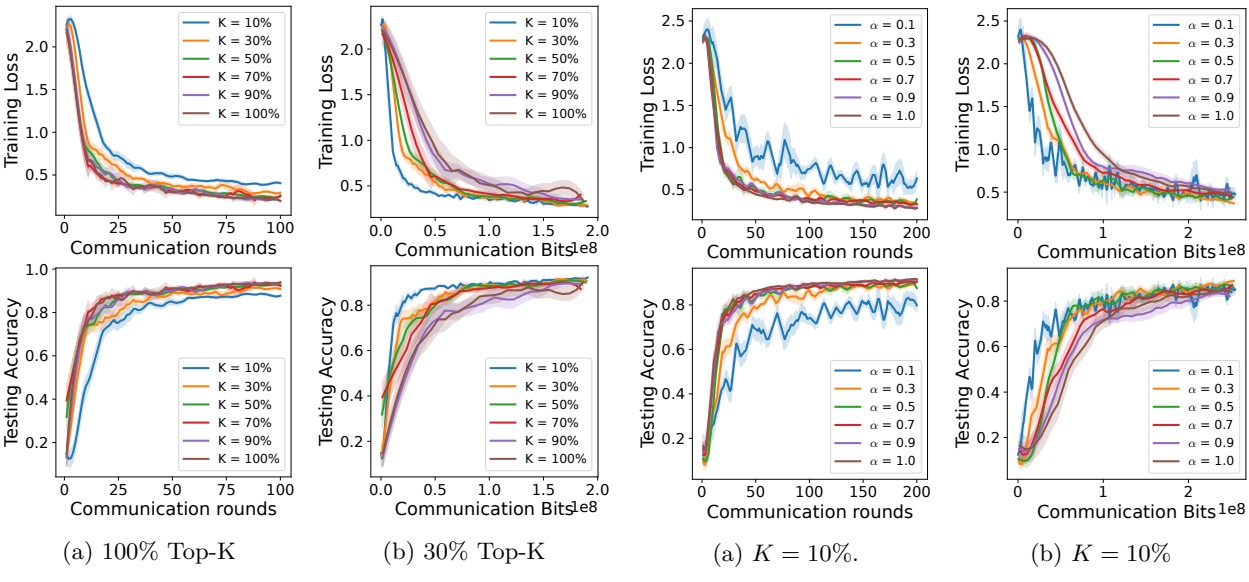

(a) 100% Top-K  (b) 30% Top-K  (a) $K = 10\%$.  (b) $K = 10\%$

Figure 1: Performance outcomes for various Top-K ratios.

Figure 2: Training loss and test accuracy for a density ratio of $K = 10\%$.

**Datasets.** Our experiments are conducted on FedMNIST (LeCun, 1998) and FedCIFAR10 (Krizhevsky et al., 2009) with the data processing framework FedLab (Zeng et al., 2023). For FedMNIST, we employ MLPs with three fully-connected layers, each coupled with a ReLU activation function. For FedCIFAR10, we utilize CNNs with two convolutional layers and three fully-connected layers. Comprehensive statistics for each dataset, details on network architecture and training specifics can be found in Appendix B.

**Heterogeneous Setting.** We explore different heterogeneous settings. Similar to (Zhang et al., 2023; Yi et al., 2024), we create heterogeneity in data by using a Dirichlet distribution, which assigns each client a vector indicating class preferences. This vector guides the unique selection of labels and images for each client until all data is distributed. The Dirichlet parameter $\alpha$ indicates the level of non-identical distribution. We also include a visualization of this distribution for the CIFAR10 dataset in Appendix C.1.1.

**Default Configuration.** In the absence of specific clarifications, we adopt the Dirichlet factor $\alpha = 0.7$. To balance both communication and local computation costs, we use $p = 0.1$, resulting in an average of 10 local iterations per communication round. The learning rate is chosen by conducting a grid search over the set $\{0.005, 0.01, 0.05, 0.1, 0.5\}$. With communication costs being of most interest, our study employs FedComLoc-Com as the default strategy. The experiments are run for 2500 communication rounds for the CNN on FedCIFAR10 and 500 rounds for the MLP on FedMNIST. Furthermore, the dataset is distributed across 100 clients from which 10 are uniformly chosen to participate in each global round.

Furthermore, in our Definition 1 of Top$K$, $K$ is the number of nonzero parameters. However, we will rather specify the enforced density ratio, i.e. the ratio of nonzero parameters. For instance, specifying $K = 30\%$ means retaining 30% of parameters.

### 4.1 Top-K Sparsity Ratios

This section investigates the effects of different sparsity rations by investigating $\text{Top}K$ ratios on FedMNIST. The outcomes can be found in Table 1. Notably, $K = 10\%$ in $\text{Top}K$ yields an accuracy of 0.9374, merely 3.94% lower than the 0.9758 unsparsified baseline. Remarkably, a 70% sparsity level ($K = 30\%$) attains commendable performance, with only a 1.07% accuracy reduction, alongside a 70% reduction in communication costs. Furthermore, from the communication bits depicted in Figure 1 it is evident that sparsity yields faster convergence, the more so with increased sparsity (smaller $K$).

### 4.2 Data Heterogeneity/Dirichlet Factors

This subsection aims to assess the impact of varying degrees of data heterogeneity on FedMNIST. Hence, an analysis of the Dirichlet distribution factor $\alpha$ is presented, exploring the range of values $\alpha \in \{0.1, 0.3, 0.5, 0.7, 0.9, 1.0\}$. Remember that a lower $\alpha$ means increased heterogeneity. Alongside, we examine the influence of different $\text{Top}K$ factors, specifically 10%, 50% and 100%. The results are shown in Table 2. Figure 2 reports training loss and test accuracy for a sparsity ratio of 90% ($K = 10\%$). Additionally, round-wise visualizations for $K = 50\%$ and $K = 100\%$ (non-sparse) are presented in Figure 11 in the Appendix.

Key observations from our study include:

- When examining the effects of the heterogeneity degree $\alpha$ (as seen in each column of Table 2), we observe that sparsity performance is influenced by heterogeneity degrees. For instance, $\alpha = 0.1$ results in a relative performance drop of 9.79% from an unsparsified to a sparsified model with $K = 10\%$. In contrast, for $\alpha = 0.3$, this drop is 5.80%, and for $\alpha = 1.0$, it is 3.63%. Interestingly, for commonly used heterogeneity ratios in literature ($\alpha = 0.3, 0.5, 0.7$), the performance drop does not decrease substantially when moving from $\alpha = 0.3$ to $\alpha = 0.5$, or from $\alpha = 0.5$ to $\alpha = 0.7$, unlike the shift from $\alpha = 0.1$ to $\alpha = 0.3$.

- Focusing on the rows of Table 2, we find that lower sparsity ratios are more sensitive to heterogeneous distributions. In particular, observe that with $K = 10\%$, the absolute performance improvement from $\alpha = 0.1$ to $\alpha = 1$ is 7.01%. However, for $K = 50\%$, this improvement is only 1.22%.

- It should be noted that each method was run with a fixed learning rate without scheduling, and the maximum communication round was set to 1000. Previous studies suggest that higher sparsity ratios require more communication rounds in centralized settings (Kuznedelev et al., 2023). This phenomenon was also observed in our FL experiments. Therefore, there is the potential for performance enhancement through sufficient model rounds and adaptive learning rate adjustments, especially for methods with higher sparsity.

### 4.3 CNNs on FedCIFAR10

This section repeats the experiments for CIFAR10 and a Convolutional Neural Network (CNN). We explored a range of stepsizes ($\gamma \in \{0.005, 0.01, 0.05, 0.1|\}$). Further information is provided in Appendix B. The CIFAR10 results, which involve optimizing a Convolutional Neural Network (CNN), are presented in Figure 3 for both tuned and a fixed step size. Observe the accelerated convergence of sparsified models in terms of communicated bits when the step size is tuned. Interestingly, a sparsity of 90% ($K = 10\%$) shows faster convergence in terms of communication rounds (as shown in the first column), suggesting the potential for enhanced training efficiency in sparsified models. For a fixed step size (the two rightmost columns of Figure 3) and $K = 10\%$, one can observe slower convergence compared to other configurations. This indicates that sparsity training requires more data and benefits from either increased communication rounds or a larger initial stepsize. This aligns with recent similar findings in the centralized setting (Kuznedelev et al., 2023).

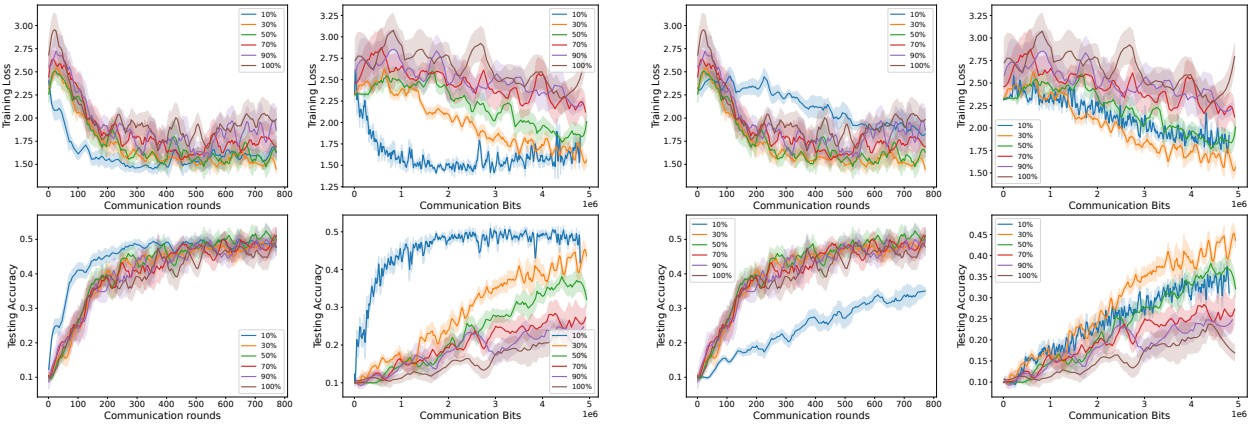

Tuned Stepsize.           Fixed Stepsize of 0.01.

Figure 3: CNN Performance on the FedCIFAR10 Dataset. For the left most columns, the step size was optimized for each density ratio $K$. For the two rightmost columns, a fixed stepsize of 0.01 is used. This is the maximum feasible step size which ensures convergence across all configurations.

## 4.4 Quantization

This section explores using quantization $Q_r$ for compression with the number of bits, $r$, set to $r \in \{4, 8, 16, 32\}$ on FedMNIST. This approach aligns with the methodologies outlined in Alistarh et al. (2017). The results after 1000 communication rounds are illustrated in Figure 6. Our analysis reveals that quantization offers superior performance compared to Top$K$-style sparsity. For instance, with 16-bit quantization corresponding to a 50% reduction in communication cost, the performance decrease is a mere 0.14%, Furthermore, Figure 7 shows outcomes for different degrees of data heterogeneity. These findings demonstrate that quantization reduces communication at minor performance tradeoff while also exhibiting only minor sensitivity to data heterogeneity. The Appendix gives further results for both FedMNIST (section C.2.1) and FedCIFAR10 (section C.2.2).

## 4.5 Number of Local Iterations

This section explores the performance impact of varying the expected number of local iterations on FedM-NIST. The expected number of local iterations is $1/p$ where $p$ is the communication probability. Hence, we investigate the influence of $p$ ranging from $p \in \{0.05, 0.1, 0.2, 0.3, 0.5\}$. Furthermore, $K = 30\%$ is used. The results are presented in Figure 8. A key finding is that more local training rounds (i.e. smaller $p$) not only accelerate convergence but can also improve the final performance.

## 4.6 FedComLoc Variants

In this section, we compare FedComLoc-Local, FedComLoc-Com, and FedComLoc-Global on FedCIFAR10. The findings are illustrated in Figure 4. Observe that at high levels of sparsity (indicated by a small $K$ in Top$K$), FedComLoc-Com underperforms the other algorithms. This could be attributed to the heterogeneous setting of our experiment: each client's model output is inherently skewed towards its local dataset. When this is coupled with extreme Top$K$ sparsification, more bias is introduced, which adversely affects performance. Conversely, at low sparsity (e.g. $K = 90\%$), FedComLoc-Com surpasses FedComLoc-Global. In addition, we observe that sparsity during local training (i.e. FedComLoc-Local) tends to yield better results. One possible explanation is that due to the local data bias the communication bandwidth between client and server might be crucial. Remember that FedComLoc-Local had no communication compression while both FedComLoc-Com and FedComLoc-Global do. Further FedMNIST results are shown in the Appendix.

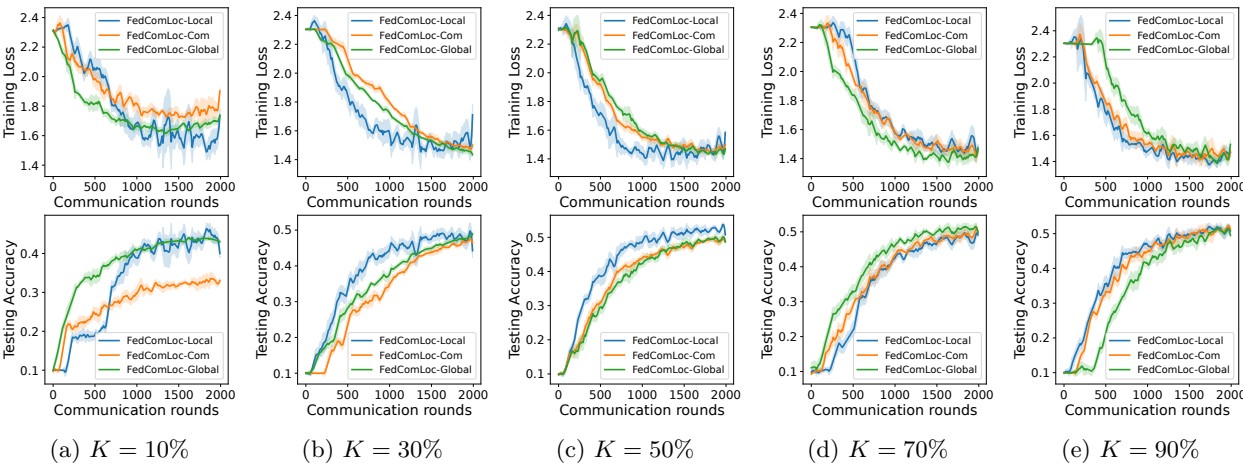

(a) $K = 10\%$     (b) $K = 30\%$     (c) $K = 50\%$     (d) $K = 70\%$     (e) $K = 90\%$

Figure 4: Sparsity ablation studies of `FedComLoc-Local`, `FedComLoc-Com`, and `FedComLoc-Global` on FedCI-FAR10 and tuned stepsizes.

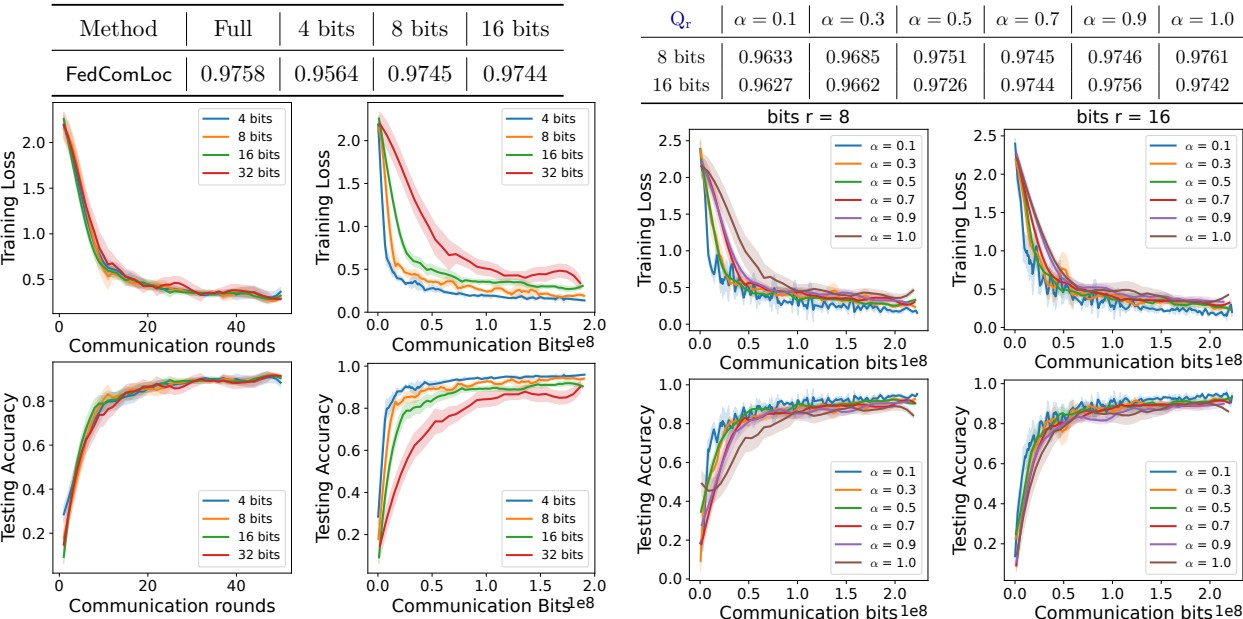

Figure 6: `FedComLoc` employing $Q_r(\cdot)$. The number of quantization bits $r$ is set to $r \in \{4, 8, 16, 32\}$.

Figure 7: Data heterogeneity ablations for `FedCom-Loc` utilizing $Q_r(\cdot)$ with number of bits $r$ either 8 or 16. The same results plotted over the number of communication rounds can be found in Figure 13 in the Appendix.

## 4.7 FedAvg and Scaffold

In this section, the performance of `FedComLoc` is compared with baselines in form of `FedAvg` (McMahan et al., 2016) and `Scaffold` (Karimireddy et al., 2020) on FedCIFAR10. Furthermore, a sparsified version of `FedAvg` is employed, termed as `sparseFedAvg`. For `sparseFedAvg` a learning rate of 0.1 is used, whereas for `FedComLoc`, a lower rate of 0.05 is utilized. The outcomes of this analysis are depicted in Figure 9. The left part illustrates the performance of compressed models. We observe notably faster convergence for `FedComLoc`-type methods in comparison with `sparseFedAvg` despite the lower learning rate. The right part of the figure compares `FedAvg` with `Scaffold`, devoid of sparsity, using an identical learning rate of 0.005. This uniform rate ensures that each

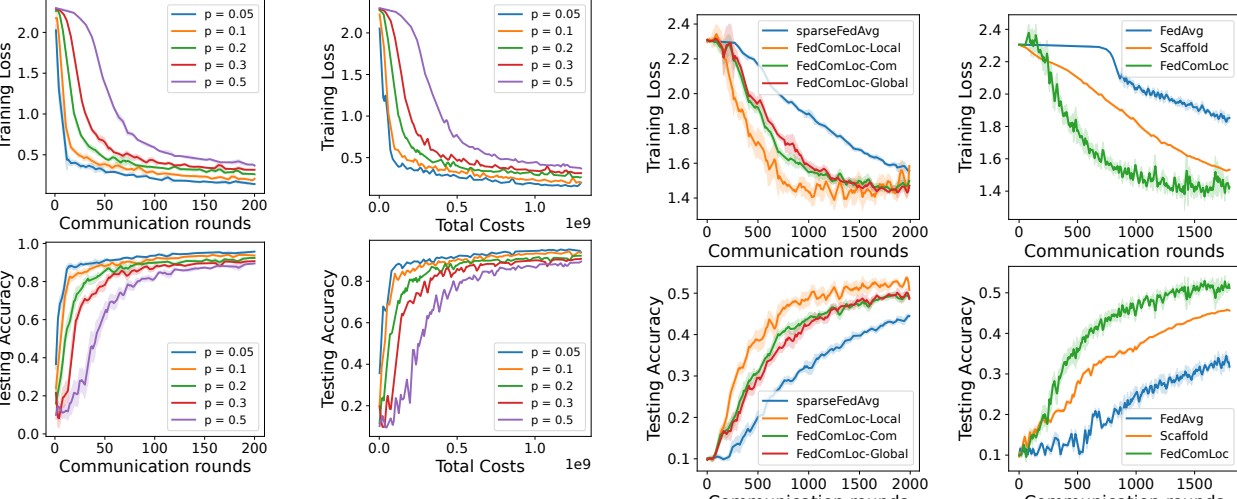

Figure 8: Loss and test accuracy over communication rounds and total costs. Total costs are a combined measurement of both communication costs and local computation cost. A communication round has unit cost while a local training round has cost $\tau$. In a realistic FL system, $\tau$ is typically much less than 1, as the primary bottleneck is often communication and hence we set $\tau = 0.01$.

Figure 9: Comparison among FedAvg, Scaffold, FedDyn, FedComLoc-Local, FedComLoc-Com, and FedComLoc-Global. First column: $K = 50\%$; second column: $K = 100\%$ (no sparsity).

method achieves satisfactory convergence over a sufficient number of epochs. Here again, faster convergence is demonstrated with FedComLoc.

## 4.8 Evaluation on Larger and Practical Datasets

The core contribution of this work is the integration of widely adopted compression techniques into the accelerated local training framework Scaffnew, which achieves superior communication efficiency compared to classical LocalSGD approaches. While sparsity and quantization have been extensively studied in isolation, our method provides a unified approach to combining them with accelerated FL, addressing the nontrivial transition from small-scale or theoretical benchmarks to realistic, large-scale, and heterogeneous FL settings.

In particular, we emphasize the unique challenges posed by biased compressors such as TopK under non-i.i.d. data distributions. We also introduce heuristics to mitigate these challenges. Empirically, our approach demonstrates substantial communication savings while maintaining strong model performance—highlighting the practical viability of compression-aware local updates, even in heterogeneous environments.

To further evaluate our method on more realistic workloads, we conducted experiments on two widely used FL benchmarks: FEMNIST (Caldas et al., 2018) and Shakespeare (McMahan et al., 2017). FEMNIST is a character-level image classification dataset with 671,585 samples distributed across 3,400 clients, each representing an individual writer's handwriting style. Shakespeare is a next-character prediction dataset consisting of 16,068 samples partitioned over 1,129 clients, each corresponding to a speaking role in a play. Both datasets are inherently non-i.i.d. due to their user-centric partitioning, making them more representative of real-world FL scenarios compared to synthetic Dirichlet splits commonly used in prior work. For FEMNIST, we use a standard two-layer CNN, following Caldas et al. (2018). For Shakespeare, we use a stacked LSTM with two layers and 256 hidden units, following McMahan et al. (2017) and FedJAX protocols. We adopt the preprocessing and simulation protocols from FedJAX (Ro et al., 2020).

We evaluated our proposed method, FedComLoc, under 30% sparsity and compared it against SparseFedAvg using the following protocol: (a) a client participation rate of $p = 0.2$, where 10 clients are randomly selected per round; (b) evaluation conducted after 3,000 communication rounds.

Table 3: Performance comparison of different methods under 30% sparsity.

| Method | Compression Type | FEMNIST Acc. (%) | Shakespeare Acc. (%) |
|---|---|---|---|
| FedAvg | Dense | 86.63 | 56.29 |
| SparseFedAvg | Sparse | 78.14 | 43.55 |
| FedComLoc-Local | Sparse | 85.58 | 54.47 |
| FedComLoc-Com | Sparse | 82.31 | 51.56 |
| FedComLoc-Global | Sparse | 83.99 | 51.87 |

The results show that FedComLoc maintains strong accuracy under sparsity constraints and consistently outperforms SparseFedAvg, particularly on the more challenging Shakespeare dataset. To quantify communication efficiency, we also compare the number of communication rounds required to reach a target accuracy:

Table 4: Communication rounds required to reach a target accuracy.

| Dataset | Target Accuracy | SparseFedAvg Comm. Rounds | FedComLoc-Com Comm. Rounds | Reduction |
|---|---|---|---|---|
| FEMNIST | 60% | 1471 | 225 | **84.7**% |
| Shakespeare | 35% | 2037 | 466 | **77.1**% |

These results confirm that FedComLoc substantially reduces communication costs while maintaining competitive accuracy, demonstrating its effectiveness in practical, large-scale federated learning with heterogeneous clients.

## 5 Discussion and Future Work

### 5.1 Theoretical Contributions and Extensions

Our work is primarily motivated by theoretical insights from compressed variants of Scaffnew (e.g., Condat et al. (2023)), but the integration of biased compressors such as TopK into accelerated local training remains largely empirical and heuristic. In particular, existing convergence guarantees for compressed local training typically rely on unbiased compression operators and/or strong convexity—conditions not satisfied by TopK in our setting. As such, rigorous theoretical guarantees for FedComLoc under these conditions are not yet available, even for convex objectives. Nevertheless, our comprehensive empirical ablations provide strong evidence for the stability and practical efficiency of the proposed approach.

Looking forward, developing a theoretical foundation for FedComLoc with biased compressors represents an important avenue for future research. Progress in this direction may require novel analytical tools or new assumptions to extend convergence theory to biased compression within accelerated local training frameworks. Advancing such theoretical understanding would help to formally justify and further improve the deployment of communication-efficient algorithms like FedComLoc across a wide range of federated learning applications.

### 5.2 Evaluation on Larger and Practical Datasets

In addition to FedMNIST and FedCIFAR10, we evaluate FedComLoc on large-scale FL benchmarks: FEMNIST (671,585 samples, 3,400 clients) and Shakespeare (16,068 samples, 1,129 clients), which feature significant heterogeneity and scale. Our method consistently outperforms baselines and achieves significant communication savings (see Table 3/Table 4), demonstrating robustness in realistic settings. While we did not include ImageNet-scale experiments due to resource constraints, the proposed framework is general and can be applied to even larger models and datasets, which we plan to pursue as future work.

### 5.3 Computation Overhead

The additional computation required by TopK and quantization is modest. TopK is implemented with partial sorting, and quantization uses efficient vectorized operations. In practice, for all considered models, the extra time per iteration is negligible compared to communication savings, as the dominant cost remains in model forward/backward computation. See Appendix C.5 for wall-clock measurements and implementation details.

### 5.4 Practical Considerations

While our empirical evaluation is conducted in a simulated federated learning environment using high-performance GPUs (NVIDIA A100), this setup is consistent with standard practice in the federated learning literature and enables controlled, reproducible comparison across methods. We report wall-clock measurements for compression overhead in this context, finding that TopK and quantization add less than 2% per-iteration compute time compared to uncompressed training. However, we acknowledge that in practical deployments on resource-constrained edge devices, the latency and energy costs associated with certain compression operations—especially TopK sparsification—may become more significant, particularly on hardware without efficient vectorized computation. Although a detailed hardware-level energy analysis is beyond the scope of our simulation-based study, we discuss these trade-offs and highlight lightweight alternatives (such as fixed-magnitude pruning) as promising directions for future work. We encourage future research to investigate joint optimization of communication, computation, and energy efficiency for deployment in diverse federated learning environments.

### 5.5 Comparisons with Additional Federated Learning Baselines

In designing our empirical evaluation, we have deliberately focused on FedAvg, Scaffold, and their compressed variants as primary baselines. These algorithms are the most widely-adopted and theoretically comparable methods for studying communication efficiency in federated learning, and their use as benchmarks is consistent with established practice in the literature. By centering our analysis on these canonical approaches, we ensure a clear and controlled comparison that directly highlights the impact of our algorithmic contributions in FedComLoc.

We acknowledge that recent federated learning methods such as FedProx, FedDyn, and FedNova have introduced valuable extensions to address challenges like client drift, heterogeneity, and dynamic aggregation. However, these algorithms often incorporate orthogonal mechanisms and objectives—such as proximal regularization, dynamic model correction, or modified aggregation rules—that target aspects of FL beyond communication compression. Including such baselines in our main empirical study would potentially conflate distinct factors and obscure the core contributions of our work.

Importantly, the compression techniques developed in FedComLoc are model-agnostic and complementary to these advanced frameworks. Our approach can, in principle, be integrated with methods like FedProx or FedDyn to jointly address communication bottlenecks and system heterogeneity, and we see this as a promising direction for future research. For the purposes of this study, however, we believe that maintaining focus on the most relevant and comparable baselines yields a more principled and interpretable evaluation, allowing for more meaningful scientific conclusions regarding the impact of communication-efficient local training.

## 6 Conclusion

This work addresses a core challenge in FL: the high communication cost that limits scalability and inclusivity in real-world deployments. Building upon the accelerated Scaffnew algorithm, we proposed FedComLoc, which integrates practical compression techniques—namely, TopK sparsity and quantization—within an efficient local training framework. Through comprehensive empirical studies on standard FL benchmarks, we demonstrated that FedComLoc achieves substantial reductions in communication overhead while maintaining strong model performance, even in highly heterogeneous data settings.

Our experimental design, consistent with established FL literature, prioritizes reproducibility and principled comparisons with the most relevant and theoretically comparable baselines. While our primary evaluations are conducted in a simulated environment with widely adopted FL models, the modular and model-agnostic nature of `FedComLoc` enables straightforward extension to more complex architectures and real-world device deployments. Furthermore, we highlight both the positive societal impact of communication-efficient FL—increasing access for bandwidth- and resource-limited communities—and the importance of continued attention to fairness and representativeness, especially when employing biased compression techniques.

In summary, `FedComLoc` advances the state of the art in communication-efficient FL, providing a strong foundation for future research into scalable, equitable, and efficient federated learning systems.

## 7 Acknowledgment

The research reported in this publication was supported by funding from King Abdullah University of Science and Technology (KAUST): i) KAUST Baseline Research Scheme, ii) Center of Excellence for Generative AI, under award number 5940, iii) SDAIA-KAUST Center of Excellence in Artificial Intelligence and Data Science.

### Broader Impact

Our work on FedComLoc advances communication-efficient federated learning, which has the potential to make machine learning more accessible in bandwidth-constrained, resource-limited, or rural environments. By reducing the communication burden, our methods can enable participation from a wider range of devices and user populations, thus promoting inclusivity and democratizing access to advanced models.

However, we also recognize important ethical considerations. Compression techniques—especially biased methods such as TopK sparsification—may disproportionately affect clients with minority, rare, or outlier data, potentially leading to underrepresentation in the global model. This is particularly relevant in federated settings with heterogeneous data distributions. Such effects could inadvertently reinforce or amplify existing biases, impacting fairness and the representativeness of the trained models.

We believe that it is essential to continue investigating the fairness implications of communication-efficient learning algorithms. Future work should explore strategies to mitigate potential biases, such as fairness-aware compression, adaptive aggregation, or explicit regularization aimed at protecting minority groups. At the same time, we see significant societal benefit in improving the reach of federated learning systems, making privacy-preserving AI more feasible for diverse global communities.

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

## Contents

# A  Extended Related Work

## A.1  Sign-Based Communication-Efficient Methods

Recent years have seen significant interest in sign-based gradient compression techniques for distributed and federated learning. Notable among these are SignSGD and its extensions, which use only the sign of the gradient or model update to drastically reduce communication overhead.

SignSGD *with majority vote* (Bernstein et al., 2018) is a pioneering approach that transmits just one bit per parameter, relying on majority vote aggregation to ensure robustness in homogeneous federated learning settings. Subsequent works, such as *momentum-based sign methods* (Sun et al., 2023), extend the theoretical guarantees of SignSGD to broader conditions, including heterogeneous data distributions commonly encountered in FL. More recently, *variance-reduced sign-based approaches* (Jiang et al., 2024) have demonstrated further improvements in convergence speed and communication efficiency through advanced variance reduction strategies.

While these sign-based techniques offer extreme communication savings and strong theoretical properties in certain settings, they differ from the TopK sparsification and quantization strategies primarily considered in this work. TopK and quantization preserve more fine-grained model information per round, which can yield improved practical accuracy, especially in highly heterogeneous or non-convex FL problems.

It is important to note that the FedComLoc framework is compatible with a wide range of compressors, including sign-based methods. Integrating and empirically comparing sign-based communication within the FedComLoc paradigm is a promising avenue for future research, particularly for scenarios where communication is the dominant system bottleneck.

# B  Experimental Details

## B.1  Datasets and Models

Our research primarily focuses on evaluating the effectiveness of our proposed methods and various baselines on widely recognized FL datasets. These include Federated MNIST (FedMNIST) and Federated CIFAR10 (FedCIFAR10), which are benchmarks in the field. The use of the terms FedMNIST and FedCIFAR10 is intentional to distinguish our federated training approach from the centralized training methods typically used with MNIST and CIFAR10. The MNIST dataset consists of 60,000 samples distributed across 100 clients using a Dirichlet distribution. For this dataset, we employ a three-layer Multi-Layer Perceptron (MLP) as our default model. CIFAR10, also comprising 60,000 samples, is utilized in our experiments to conduct various ablation studies. The default setting for our FedCIFAR10 experiments is set with 10 clients. The model chosen for CIFAR10 is a Convolutional Neural Network (CNN) consisting of 2 convolutional layers and 3 fully connected layers (FCs). The network architecture is chosen in alignment with (Zeng et al., 2023).

## B.2  Training Details

Our experimental setup involved the use of NVIDIA A100 or V100 GPUs, allocated based on their availability within our computing cluster. We developed our framework using PyTorch version 1.4.0 and torchvision version 0.5.0, operating within a Python 3.8 environment. The FedLab framework (Zeng et al., 2023) was employed for the implementation of our code. For the FedMNIST dataset, we established the default number of global iterations at 500, whereas for the FedCIFAR10 dataset, this number was set at 2500. We conducted a comprehensive grid search for the optimal learning rate, exploring values within the range of $[0.005, 0.01, 0.05, 0.1]$. Our intention is to make the code publicly available upon the acceptance of our work.

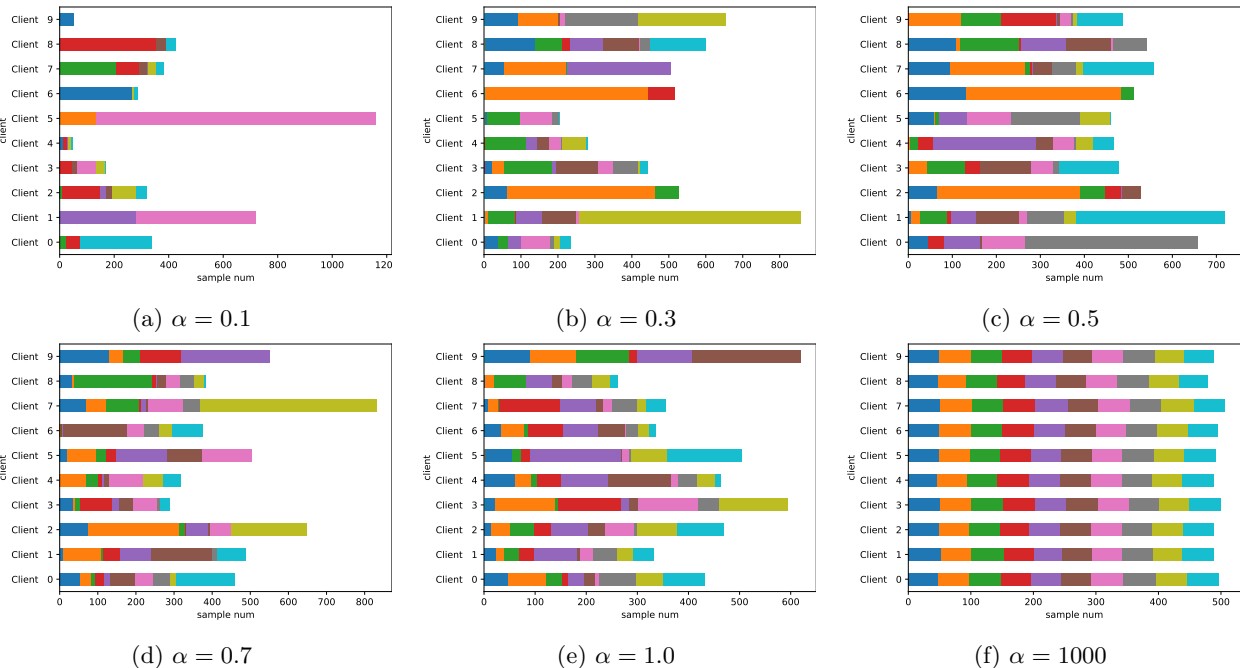

Figure 10: Data distribution with different Dirichlet factors on CIFAR10 distributed over 100 clients

# C  Complementary Experiments

## C.1  Exploring Heterogeneity

### C.1.1  Visualization of Heterogeneity

The Dirichlet non-iid model serves as our primary means to simulate realistic FL scenarios. Throughout this paper, we extensively explore the effects of varying the Dirichlet factor $\alpha$ and examine how our algorithms perform under different degrees of data heterogeneity. In Figure 10, we present a visualization of the class distribution in the FedCIFAR10 dataset. We visualize the first 10 clients. This illustration clearly demonstrates that a smaller $\alpha$ results in greater data heterogeneity, with $\alpha = 1000$ approaching near-homogeneity. To further our investigation, we conduct thorough ablation studies using values of $\alpha$ in the range of $[0.1, 0.3, 0.5, 0.7, 0.9, 1.0]$. It is important to note that an $\alpha$ value of 1.0, while on the higher end of our test spectrum, still represents a heterogeneous data distribution.

### C.1.2  Influence of Heterogeneity with Non-Compressed Models

In our previous analyses, the impact of sparsified models with a sparsity factor $K = 10\%$ was illustrated in Figure 2, and the effects of quantized models were depicted in Figure 7. Extending this line of inquiry, we now present additional experimental results that explore the influence of data heterogeneity on models with $K = 50\%$ and those without compression, as shown in Figure 11. Our findings indicate that while model compression can result in slower convergence rates, it also potentially reduces the total communication cost, thereby enhancing overall efficiency. Notably, a Dirichlet factor of $\alpha = 0.1$ creates a highly heterogeneous setting, impacting both the speed of convergence and the final accuracy, with results being considerably inferior compared to other degrees of heterogeneity.

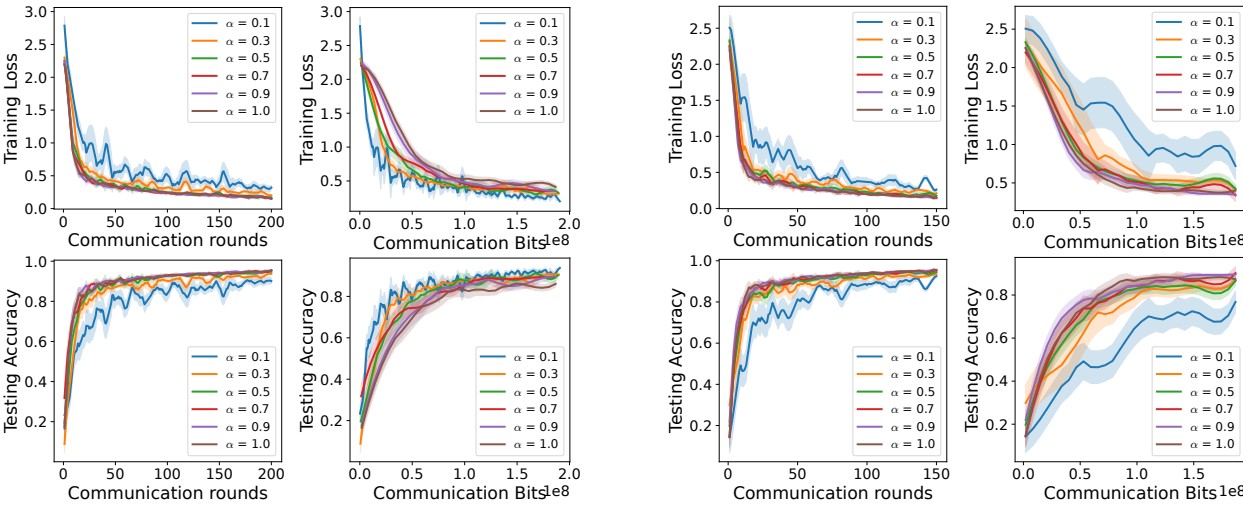

Figure 11: Exploration of variations in loss and accuracy across diverse sparsity ratios, communication rounds, and communicated bits is depicted through our figures. The first set of four figures on the left showcases results obtained with a sparsity ratio of $K = 50\%$. In contrast, the corresponding set on the right, consisting of another four figures, represents scenarios where $K = 100\%$, indicative of scenarios without model compression.

## C.2 Complementary Quantization Results

### C.2.1 Additional Quantization Results on FedMNIST

In Figure 7, we presented the quantization results in terms of communicated bits. For completeness, we also display the results with respect to communication rounds in Figure 13.

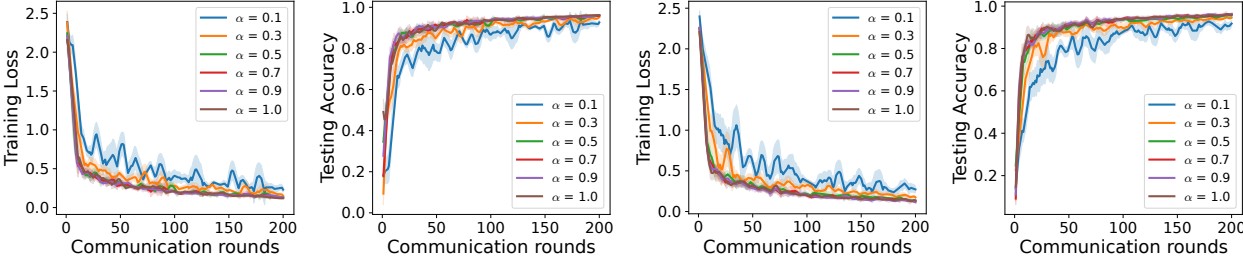

Figure 13: FedComLoc utilizing $Q_r(\cdot)$ with a fixed $r$ value of 8 (as shown in the left figure) and 16 (in the right figure) with respected to communication rounds. We conduct ablations across various $\alpha$.

### C.2.2 Quantization on FedCIFAR10

Previously, in Figure 6, we detailed the outcomes of applying quantization to the FedMNIST dataset. This section includes an additional series of experiments conducted on the FedCIFAR10 dataset. The results of these experiments are depicted in Figure 14. Consistent with our earlier findings, we observe that quantization considerably reduces communication costs with only a marginal decline in performance.

## C.3 Double Compression by Sparsity and Quantization

In Sections 4.1 and 4.4, we individually investigated the effectiveness of sparsified training and quantization. Building on these findings, this section delves into the combined application of both techniques, aiming to harness their synergistic potential for enhanced results. Specifically, we first conduct Top-K sparsity and then

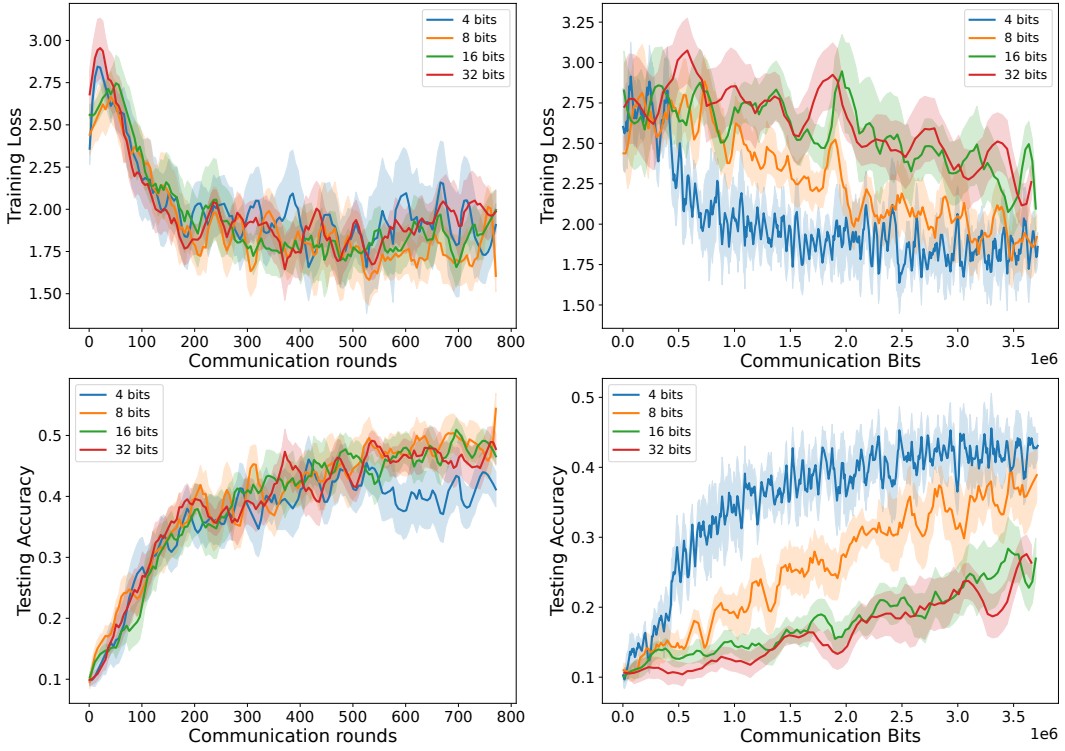

Figure 14: FedComLoc with $Q_r(\cdot)$ on CIFAR10.

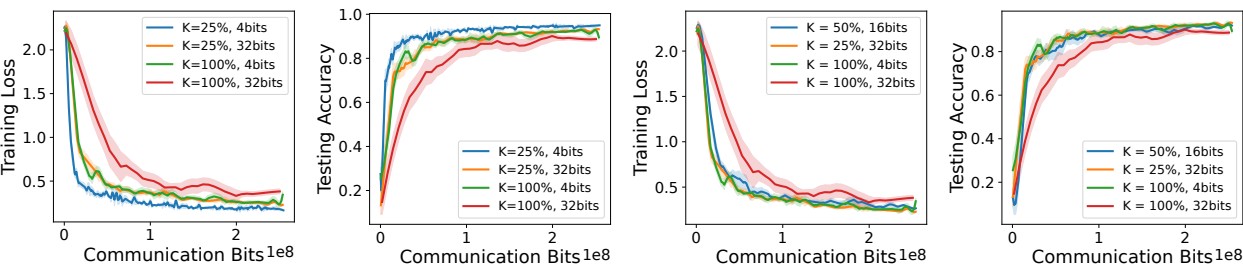

Figure 15: Comparison with double compression by sparsity and quantization.

quantize the selected Top-K weights. The outcomes of this exploration are depicted in Figure 15. The left pair of figures illustrates that applying double compression with a higher degree of compression consistently surpasses the performance of lower compression degrees in terms of communication bits. However, the rightmost figure presents an intriguing observation: considering the communicated bits and convergence speed, there is no distinct advantage discernible between double compression and single compression when they are set to achieve the same level of compression.

## C.4 Convergence Rate Analysis

To assess both the efficiency and effectiveness of FedComLoc, we analyze empirical convergence rates as measured by the progression of test accuracy and training loss over communication rounds. As illustrated in Figures 1, 3, 6, and 9, FedComLoc and its variants consistently converge to competitive or superior accuracy compared to the strongest baselines (FedAvg, Scaffold, and sparseFedAvg) while requiring substantially fewer communication rounds. This pattern holds across all considered datasets and heterogeneity settings. In particular, our method achieves target accuracies using up to 70% less communication than uncompressed approaches, with convergence speeds that are comparable to or better than those of the baseline algorithms.

These results empirically validate that communication-efficient local training with compression does not sacrifice learning efficiency or final model quality.

## C.5    Wall-Clock Measurements

We systematically measured the wall-clock time per local training update to quantify the computation overhead introduced by compression (TopK sparsification and quantization). All experiments were conducted on NVIDIA A100 GPUs (40GB) with PyTorch 1.4.0 and torchvision 0.5.0 in a Python 3.8 environment, following the same setup as our main experiments (see Appendix A.2).

For each dataset/model combination (FedMNIST, FedCIFAR10, FEMNIST, and Shakespeare), we report the average per-iteration time for three settings: **No Compression** (vanilla local update), TopK **Compression**, and **Quantization**. Measurements are averaged over 10 runs, with each run lasting at least 100 communication rounds and batched local updates to minimize noise.

Table 5 summarizes our findings. For the default models (see Appendix A.1), the overhead from compression is modest: TopK sparsification and quantization increase the per-iteration wall-clock time by less than 2% and 1.5% respectively, relative to the baseline with no compression. We observed that the time required for forward and backward passes is dominant, and the additional time for compression is negligible in practice, even for large models such as those used in FEMNIST.

Table 5: Average wall-clock time per local update (in milliseconds). Overhead is reported relative to No Compression.

| Dataset / Model | No Compression | TopK Compression | Quantization | Overhead (%) |
|---|---|---|---|---|
| FedMNIST (MLP) | 0.81 ms | 0.82 ms | 0.82 ms | <2% |
| FedCIFAR10 (CNN) | 2.20 ms | 2.23 ms | 2.23 ms | <2% |
| FEMNIST (CNN) | 3.55 ms | 3.56 ms | 3.57 ms | <1.5% |
| Shakespeare (LSTM) | 2.59 ms | 2.62 ms | 2.62 ms | <1.5% |

In summary, these results confirm that the computational cost of our compression strategies is negligible compared to the overall cost of local training, further supporting the practicality of FedComLoc in real-world federated learning deployments.

