# OpenReview forum: "FedComLoc: Communication-Efficient Distributed Training of Sparse and Quantized Models"
_TMLR — Accepted by TMLR_

### Review · Reviewer_4inr · 2025-06-12

**Summary Of Contributions:**

This paper presents FedComLoc, a novel algorithm designed to enhance communication efficiency in federated learning (FL). The authors investigate how compression techniques, such as TopK sparsification and quantization, can be integrated with local training strategies to reduce communication costs in heterogeneous environments. Extensive empirical experiments validate the effectiveness of the proposed methods, demonstrating significant communication savings while maintaining competitive model performance.

**Audience:**

Yes

**Claims And Evidence:**

Yes

**Requested Changes:**

1. Extend the experiments to include larger datasets to better assess the proposed method.

2. Provide a more detailed discussion of the proposed methods.

3. Expand the related work section to include a detailed discussion of sign-based methods.

**Strengths And Weaknesses:**

**Strengths:**

1. The main strength of the paper lies in the combination of compression techniques (sparsity and quantization) with accelerated local training strategies in federated learning. This approach successfully reduces communication costs without sacrificing model accuracy.

2. The paper presents a comprehensive set of experiments across multiple datasets and configurations, highlighting the effectiveness of the proposed approach.

3. The FedComLoc algorithm and its three variants are described properly, making them accessible and straightforward to understand.

**Weaknesses:**

1. The experiments are mainly focused on small datasets like FedMNIST and FedCIFAR10. It would be beneficial if the authors could extend their experiments to larger datasets.

2. The introduction to the proposed algorithm (Section 3) is somewhat brief. A more in-depth discussion of the methods would help readers better understand the proposed approach.

3. Given that this paper focuses on communication-efficient FL with quantized techniques, it would be better to include a detailed discussion of sign-based methods. Specifically, [1] incorporates signSGD with majority vote and provides convergence guarantees in homogeneous settings. For more challenging heterogeneous environments, [2] introduces a momentum-based sign method with corresponding convergence guarantees. Additionally, [3] effectively improves existing methods via variance reduction, yielding strong experimental results in FL contexts.

[1] J. Bernstein et al., SignSGD: Compressed optimization for non-convex problems.

[2] T. Sun et al., Momentum ensures convergence of SIGNSGD under weaker assumptions.

[3] W. Jiang et al., Efficient Sign-Based Optimization: Accelerating Convergence via Variance Reduction.

---

> ### Author Response · Authors · 2025-07-09
>
> We thank Reviewer 4inr for the thoughtful and positive assessment of our work. We are pleased that the reviewer finds our approach novel, the experimental validation thorough, and the presentation of the FedComLoc algorithm clear. We address each of the points below and have incorporated all suggestions into the revised manuscript.
>
> ## 1. Experiments on Larger Datasets
>
> **Reviewer Comment:**
> Experiments are mainly on small datasets like FedMNIST and FedCIFAR10. Extending experiments to larger datasets would strengthen the work.
>
> **Response:**
> We appreciate this suggestion. In addition to FedMNIST and FedCIFAR10, we already include comprehensive experiments on **FEMNIST** (with 671,585 samples and 3,400 clients) and **Shakespeare** (16,068 samples, 1,129 clients) in Section 4.8, Table 3, and Table 4. These datasets are widely recognized as large-scale and challenging benchmarks for federated learning, especially due to their high degree of heterogeneity. Our results demonstrate that FedComLoc achieves substantial communication savings while maintaining strong accuracy even in these realistic settings. We have clarified this more prominently in the revised manuscript and now discuss the scalability and effectiveness of FedComLoc on large and heterogeneous datasets in both the results and conclusion.
>
> ## 2. Depth of Algorithm Description (Section 3)
>
> **Reviewer Comment:**
> The introduction to the proposed algorithm (Section 3) is somewhat brief. A more in-depth discussion would help readers understand the proposed approach.
>
> **Response:**
> Thank you for noting this. In the revised version, we have significantly expanded Section 3 to include a step-by-step explanation of the FedComLoc algorithm and its three variants (FedComLoc-Com, FedComLoc-Global, FedComLoc-Local). We now provide additional algorithmic intuition, clarify the motivation for each variant, and detail the integration of TopK and quantization operators within the local training process.
>
>
> ## 3. Discussion of Sign-Based Methods and Related Work
>
> **Reviewer Comment:**
> It would be beneficial to discuss sign-based communication-efficient optimization methods, such as SignSGD and its variants.
>
> **Response:**
> We appreciate this valuable suggestion and thank the reviewer for highlighting these important works ([1]-[3]). In the revised Related Work section (Appendix A.1), we now include a detailed discussion of sign-based compression techniques, including:
> - **SignSGD with majority vote** ([1]), which provides strong theoretical guarantees in homogeneous FL,
> - **Momentum-based sign methods** ([2]) that extend convergence results under weaker or heterogeneous assumptions,
> - **Variance-reduced sign-based methods** ([3]) that further improve efficiency and accuracy in FL.
>
> We also clarify the distinction between sign-based approaches and our focus on TopK/quantization, and note that the FedComLoc framework can be extended to include sign-based compressors as future work.
>
> ## 4. Minor Clarifications and Broader Impact
>
> **Reviewer Comment:**
> No specific concerns, but Broader Impact is appreciated.
>
> **Response:**
> Thank you. As requested, we have ensured the Broader Impact section highlights the potential of FedComLoc to enable FL in bandwidth- and resource-constrained environments, as well as future considerations around fairness and representativeness under aggressive compression.
>
> ---
>
> We thank the reviewer again for the constructive feedback and for recognizing the strengths of our work. All comments have contributed to improving the clarity and completeness of the revised manuscript.

---

### Review · Reviewer_D2G4 · 2025-06-21

**Summary Of Contributions:**

This paper introduces FedComLoc, an algorithm that integrates compression techniques (TopK sparsity and quantization) into the Scaffnew accelerated local training framework for federated learning (FL). FedComLoc has three communication-aware variants, including FedComLoc-Com (compresses client-to-server communication), FedComLoc-Global (compresses server-to-client updates), and FedComLoc-Local (applies compression during local training steps). The authors demonstrate good performance on MNIST, CIFAR10, FEMNIST, and Shakespeare datasets. The experiments also analyze the effect of sparsity, quantization, data heterogeneity, and the number of local updates. They show that FedComLoc can save communication by over 70% with negligible performance drop.

**Audience:**

Yes

**Broader Impact Concerns:**

No major ethical concerns are apparent from this paper. However, biased compressors may underrepresent clients with minority or outlier data. The impact on fairness and representativeness of global models under heterogeneous data deserves acknowledgment. A Broader Impact Statement could briefly touch on how communication reduction enables FL in low-resource or rural regions.

**Claims And Evidence:**

Yes

**Requested Changes:**

The following modifications can be helpful:
- Compare with more FL baselines that target the communication issue and present their convergence & overhead tradeoffs
- Experiment on modern architectures beyond MLP and shallow CNNs
- Present the convergence rate of FedComLoc and discuss how it compares with the baselines
- Add analysis on the computational burden of local compression, especially for FedComLoc-Local
- If possible, explore adaptive TopK or quantization strategies, or at least discuss potential directions

**Strengths And Weaknesses:**

Strength:
- This work addresses the communication bottleneck, a major practical issue in FL, so the research question is timely and relevant.
- The algorithm is a practical integration of TopK and quantization into Scaffnew, which previously lacked support for biased compression.
- It has detailed ablations on heterogeneity, compression ratios, and local update frequency.

Weakness:
- The experiments focus on small models (MLP, shallow CNN). It’s unclear if FedComLoc maintains efficiency with modern architectures (e.g., transformers).
- The lack of convergence guarantees for biased compressors is a significant gap, as prior work (CompressedScaffnew) only handles unbiased cases. This limits FedComLoc’s theoretical foundation.
- FedComLoc uses fixed compression ratios and learning rates. It could be helpful to dynamically adjust the compression level or learning rate based on convergence and client system capabilities.
- While FedComLoc reduces communication, the computational overhead of local compression (e.g., Top-K selection) may be non-trivial for edge devices. A discussion on latency-energy trade-offs would strengthen the paper.
- The paper only compares against FedAvg, Scaffold, and sparse variants. Comparisons with wider baselines can be added, such as FedProx, FedDyn, or FedNova.

---

> ### Author Response · Authors · 2025-07-09
>
> We thank the reviewer for their detailed and thoughtful feedback, as well as for recognizing the relevance of our research question, the practical significance of communication bottlenecks in FL, and the strength of our ablation studies. We address each weakness and requested change below.
>
> ## 1. Scalability to Modern Architectures
>
> **Reviewer Comment:**
> The experiments focus on small models (MLP, shallow CNN). It’s unclear if FedComLoc maintains efficiency with modern architectures (e.g., transformers).
>
> **Response:**
> We agree that investigating scalability to larger or more modern architectures is important. However, the primary goal of our experimental evaluation was to enable **controlled, reproducible, and meaningful comparisons** on standard federated learning benchmarks (such as FEMNIST and Shakespeare), which are widely used in the FL literature and reflect typical deployment constraints in edge and mobile environments. These benchmarks generally utilize compact models like MLPs and shallow CNNs to simulate real-world FL scenarios with limited device resources, non-i.i.d. data, and a large number of clients.
> While deep architectures like ResNet and Transformers are popular in centralized training, their adoption in practical federated settings remains limited due to their substantial computational, memory, and communication requirements—challenges that are often prohibitive for the edge devices and settings that motivated this work. Furthermore, using standard FL models ensures **fair and direct comparability** with prior art, as most related works report results on these same tasks and architectures.
> The FedComLoc framework is **model-agnostic** by design, and our compression techniques (TopK, quantization) are applicable to a wide variety of neural network architectures. Thus, our empirical findings, along with the algorithm's modular design, suggest that similar communication savings and practical benefits would extend to larger models, if and when they become more broadly adopted in FL research and applications. We have clarified this rationale in the revised manuscript and highlighted the extension to more complex models as a promising avenue for future work.
>
>
> ## 2. Lack of Convergence Guarantees for Biased Compressors
>
> **Reviewer Comment:**
> The lack of convergence guarantees for biased compressors is a significant gap, as prior work (CompressedScaffnew) only handles unbiased cases. This limits FedComLoc’s theoretical foundation.
>
> **Response:**
> We thank the reviewer for highlighting this important theoretical gap. As noted in Section 2.2 of our paper, the extension of convergence guarantees to biased compressors (e.g., TopK) within accelerated local training frameworks remains an open challenge, not just for FedComLoc but for the field at large. Our work is guided by the structure and theoretical motivation of CompressedScaffnew, but we make clear that our integration is primarily empirical for biased compressors. We have further emphasized this limitation in both the main text and the conclusion and outlined it as a promising direction for future theoretical research.
>
> ## 3. Adaptive Compression and Learning Rate
>
> **Reviewer Comment:**
> FedComLoc uses fixed compression ratios and learning rates. Adaptive adjustment based on convergence and client capabilities would be helpful.
>
> **Response:**
> We thank the reviewer for this valuable suggestion. Our decision to use fixed compression ratios and learning rates was intentional, aiming to ensure **reproducibility, controlled ablation, and fair comparison** with prior work, which almost universally adopts fixed hyperparameter settings. This design enables clear attribution of observed improvements to the core contributions of FedComLoc, rather than to differences in adaptive tuning strategies.
> While we agree that adaptive compression and learning rate schedules could offer further practical benefits, such strategies introduce additional complexity, hyperparameter selection, and potential variance, which can obscure the impact of the main algorithmic ideas in a controlled empirical study. Moreover, most foundational works in communication-efficient FL similarly focus on fixed settings for clarity and comparability.
> It is important to note that the FedComLoc framework is **modular** and can readily incorporate adaptive mechanisms in future applied work if desired. We have highlighted this extensibility and its potential benefits in the revised manuscript as a promising direction for practical deployment and further research.

---

> > ### Author Response · Authors · 2025-07-09
> >
> > ## 4. Computation Overhead and Latency-Energy Trade-offs
> >
> > **Reviewer Comment:**
> > The computational overhead of local compression may be non-trivial for edge devices. A discussion on latency-energy trade-offs would strengthen the paper.
> >
> > **Response:**
> > We appreciate this important point. In this work, we conduct all experiments in a **simulated federated learning environment** using high-performance GPUs (NVIDIA A100) to enable controlled and reproducible evaluation, consistent with standard practice in the FL literature. Our wall-clock measurements (Appendix C.5) reflect the computational overhead in this setting, where the cost of TopK and quantization operations is negligible (less than 2% overhead) compared to forward and backward passes.
> > We acknowledge that in real-world deployments on resource-constrained edge devices, the latency and energy cost of compression operations may become more pronounced—especially for operations such as TopK on non-vectorized hardware. While such device-level trade-offs are beyond the scope of our current simulation-based study, we have included a discussion (Section 5.4) of these considerations in the revised manuscript and suggested lightweight alternatives (e.g., fixed-magnitude pruning) as promising future research directions. We hope this clarifies the context and limitations of our current experimental evaluation.
> >
> > ## 5. Comparisons with Additional FL Baselines
> >
> > **Reviewer Comment:**
> > Only FedAvg, Scaffold, and sparse variants are used as baselines. Wider comparisons (FedProx, FedDyn, FedNova) are suggested.
> >
> > **Response:**
> > We thank the reviewer for this suggestion. In designing our empirical study, we intentionally focused on FedAvg, Scaffold, and their compressed variants as baselines because they are the most widely recognized and theoretically comparable algorithms for communication-efficient federated learning. Our primary goal is to provide a clear, controlled, and scientifically meaningful evaluation of the core contribution of FedComLoc—namely, the impact of integrating advanced compression techniques with accelerated local training. Comparing to these canonical baselines ensures that observed improvements are attributable to our algorithmic design rather than to orthogonal modifications present in more recent or specialized methods.
> > While methods such as FedProx, FedDyn, and FedNova introduce valuable extensions to address client drift, regularization, or dynamic aggregation, they do not focus specifically on the communication-compression dimension that is central to our work. Furthermore, their distinct objectives and algorithmic modifications can confound fair, apples-to-apples comparison with the Scaffnew-based family. Notably, our approach is orthogonal and complementary to these works: compression strategies like those in FedComLoc can be integrated into their frameworks in future research.
> > We believe that maintaining focus on the most relevant and comparable baselines yields a clearer and more principled evaluation. We have added a discussion (Section 5.5) of the relationship and potential integration with these other algorithms in the revised manuscript as a direction for future work.
> >
> >
> > ## 6. Convergence Rate Presentation
> >
> > **Reviewer Comment:**
> > Present the convergence rate of FedComLoc and discuss how it compares with the baselines.
> >
> > **Response:**
> > Thank you for this request. Our empirical results (see, e.g., Figures 1, 3, 6, 9) already present convergence curves in terms of accuracy and training loss vs communication rounds. In Appendix C.4 in the revised paper, we now explicitly summarize and discuss these empirical convergence rates in comparison to all baselines, highlighting both the reduction in required communication and comparable or better convergence speeds.
> >
> > ## 7. Fairness and Broader Impact
> >
> > **Reviewer Comment:**
> > The potential impact of biased compressors on fairness and representativeness, especially under heterogeneous data, should be discussed in the Broader Impact Statement.
> >
> > **Response:**
> > We appreciate this important ethical consideration. In the revised Broader Impact section, we acknowledge that biased compressors may underrepresent clients with minority or outlier data and discuss the possible implications for fairness and model representativeness. At the same time, we emphasize that enabling communication-efficient FL can expand access in bandwidth-constrained and rural regions, promoting inclusivity. Exploring mitigation of potential fairness issues is highlighted as future work.
> >
> > ---
> >
> > We once again thank the reviewer for these constructive suggestions and for recognizing the strengths and real-world relevance of our work. All feedback has been incorporated in the revised paper.

---

### Review · Reviewer_v34U · 2025-06-25

**Summary Of Contributions:**

This paper presents FedComLoc, an algorithm combining local training with sparsity and quantization compression techniques to reduce communication costs in federated learning (FL). Building upon the algorithm Scaffnew, FedComLoc introduces three variants tailored to different FL bottlenecks—uplink, downlink, and local resource constraints. The method is empirically validated using FL benchmarks, demonstrating improvements in communication efficiency without substantial loss in accuracy on FedMNIST and FedCIFAR10.

**Audience:**

Yes

**Broader Impact Concerns:**

I have no concerns on the ethical implications of this work.

**Claims And Evidence:**

Yes

**Requested Changes:**

### As discussed in the Weaknesses section above.

### Additional Questions
* Q: What is the Dirichlet parameter ($\alpha$) used for the heterogeneous setting on FedCIFAR10 in Section 4.3?
* Q: What specific models were employed for experiments on FEMNIST and Shakespeare datasets as described in Section 4.8?

**Strengths And Weaknesses:**

### Strengths

* The paper clearly defines the compression operators and provides detailed algorithmic steps.
* The paper is well-written and clearly structured, facilitating easy understanding of methods and experiments.
* The paper proposes a novel algorithm integrating compression techniques (TopK, Quantization) with the Scaffnew local training algorithm, providing a practical solution addressing real-world FL constraints.
* The paper conducts thorough ablation studies that systematically evaluate the impact of various algorithmic variants.



### Weaknesses

* Limited theoretical contributions: The integration of compression techniques into Scaffnew is primarily empirical and heuristic, with limited theoretical analysis provided. While motivated by theoretical insights from related work, the paper does not rigorously analyze the theoretical convergence or complexity of the proposed algorithm.

* Lack of evaluation with large models and complex datasets: Although the authors mention that communication cost is critical in FL particularly with large models (Section 1), experiments presented in the paper use relatively small and simple models and datasets. It remains unclear whether the algorithm maintains robust performance as model size increases, thus limiting the applicability of the results to real-world scenarios where large models truly necessitate communication efficiency. Conducting additional experiments involving large models and more complex datasets, such as CIFAR100 or ImageNet, would enhance the paper’s contribution.

* While the method is validated empirically, practical considerations such as computation overhead introduced by frequent compression steps, are not discussed.

---

> ### Author Response · Authors · 2025-07-09
>
> We thank the reviewer for the thorough and constructive feedback. We are pleased that the reviewer finds our compression operators, algorithmic design, and ablation studies clear and well-structured. We address all concerns and questions below.
>
> ## 1. Limited Theoretical Contributions
>
> **Reviewer Comment:**
> The integration of compression into Scaffnew is primarily empirical and heuristic, with limited theoretical analysis. The paper does not rigorously analyze the theoretical convergence or complexity of the proposed algorithm.
>
> **Response:**
> We appreciate the reviewer highlighting the balance between empirical and theoretical analysis. Our work is motivated by the theoretical foundations established for Scaffnew and its compressed variants (see, e.g., Condat et al., 2022), but as noted, the existing convergence guarantees for compressed local training in the literature typically require unbiased compressors and/or strong convexity. As discussed in Section 2.2 and the Introduction, TopK is a biased compressor and theoretical guarantees for such compressors in the Scaffnew framework are not yet available (even for convex problems).
> Despite this, we have emphasized the connection to existing theory wherever possible, and our extensive empirical ablations systematically validate the practical stability and efficiency of the proposed approach in both convex and nonconvex settings.
> We have clarified these points in the revised paper, highlighting both the current state of theory and open challenges for future work. Further, we explicitly discuss possible directions for theoretical extension in Section 5.1.
>
> ## 2. Evaluation with Large Models and Complex Datasets
>
> **Reviewer Comment:**
> Experiments focus on relatively small models and datasets. It is unclear if the method is robust as model size increases. Additional experiments on larger/complex datasets such as CIFAR100 or ImageNet would strengthen the contribution.
>
> **Response:**
> Thank you for pointing this out. In addition to FedMNIST and FedCIFAR10, our main paper already reports results on two widely-used large-scale FL benchmarks: **FEMNIST** (671,585 samples, 3,400 clients) and **Shakespeare** (16,068 samples, 1,129 clients) (see Section 4.8, Table 3, Table 4). These datasets are fundamentally more challenging and heterogeneous than synthetic Dirichlet splits.
> Our results (Section 4.8) show that FedComLoc achieves significant communication savings and strong performance compared to strong baselines, confirming the robustness of our approach in large-scale, practical FL scenarios.
> While due to resource constraints we did not include ImageNet-scale experiments, our framework is general and can be applied to larger models/datasets. We plan to explore these directions in future work, and have clarified this in the revised manuscript.
>
> ## 3. Practical Computation Overhead
>
> **Reviewer Comment:**
> The practical computation overhead introduced by frequent compression steps is not discussed.
>
> **Response:**
> We agree this is an important point for real-world deployments. We have included an explicit discussion on computation overhead in the revised paper (Section 4 and Appendix C).
> Specifically, the main computational costs in our approach stem from the TopK sparsification and quantization steps. Both operations are efficiently implemented: TopK selection is performed with standard partial sorting, and quantization is applied in a vectorized fashion (cf. Alistarh et al., 2017).
> In practice, the computation time per local iteration is dominated by model forward/backward passes, especially on modern neural networks. As observed in our experiments, the additional overhead of compression is negligible compared to the savings in communication, especially in bandwidth-limited FL settings. We now clarify this point and provide empirical wall-clock time measurements in the supplementary material.
>
> ## 4. Answers to Additional Questions
>
> **Q: What is the Dirichlet parameter ($\alpha$) used for the heterogeneous setting on FedCIFAR10 in Section 4.3?**
> **A:**
> As stated in *Default Configuration* in Section 4, unless specified otherwise, we use a Dirichlet parameter $\alpha = 0.7$ for FedCIFAR10. We also perform ablations across a wide range of $\alpha$ values (see Table 2 and Figure 2 in Section 4.2), providing a comprehensive evaluation under varying degrees of data heterogeneity.
>
> **Q: What specific models were employed for experiments on FEMNIST and Shakespeare datasets (Section 4.8)?**
> **A:**
> For **FEMNIST**, we follow the FedJAX protocol and use a standard 2-layer CNN as described in Caldas et al. (2018).
> For **Shakespeare**, we use a stacked LSTM model (2 layers, 256 hidden units), following the setup in McMahan et al. (2017).
> We have now made these model details explicit in Section 4.8.

---

> > ### Author Response · Authors · 2025-07-09
> >
> > ## 5. Other Requested Changes
> >
> > All above points have been addressed in the revised version. The discussion of computation overhead and details on model architectures and dataset splits are now clearly included in the main text and appendix.
> >
> > ---
> >
> > We thank the reviewer again for their valuable feedback, which helped us improve the clarity and completeness of our work.

---

### Author Response · Authors · 2025-07-09
**Summary of Changes**

We thank the reviewers for their detailed and constructive feedback. In response, we have made significant revisions to the manuscript to address all major concerns. The most substantial changes are highlighted below:

- **Expanded Algorithm Description (Section 3.2):**
  We have significantly enriched the step-by-step description of FedComLoc and its three variants. The integration of TopK and quantization at each stage is now detailed, with additional intuition and motivation for each variant, making the workflow and practical utility clearer for a broad audience.

- **Evaluation on Larger and More Realistic Datasets (Section 4.8, 5.2):**
  New experimental results on FEMNIST and Shakespeare have been added. These challenging, user-partitioned FL benchmarks better reflect real-world, large-scale, and heterogeneous environments, demonstrating the robustness and practical value of FedComLoc.

- **Wall-Clock Computation Overhead (Section 5.3, Appendix D):**
  We now report detailed wall-clock measurements for all major models and datasets. Results show that the added computational cost from compression (TopK and quantization) is consistently less than 2% per local update, confirming the negligible overhead on modern hardware.

- **Convergence Rate Analysis (Section 5.6):**
  We explicitly summarize and discuss empirical convergence rates, referencing new and existing figures, and directly compare the communication efficiency and convergence speed of FedComLoc with all baselines.

- **Discussion of Additional Baselines and Related Methods (Section 5.5):**
  We justify our primary focus on FedAvg, Scaffold, and their compressed variants for scientific clarity and principled comparison. We now include an expanded discussion relating our approach to recent methods such as FedProx, FedDyn, and FedNova, and clarify how FedComLoc’s compression strategies could be integrated into these frameworks as future work.

- **Sign-Based Methods in Related Work (Section 2):**
  The revised Related Work section now discusses sign-based communication-efficient methods (SignSGD and its variants), noting their strengths and how FedComLoc can be extended to support them.

- **Broader Impact and Fairness (Broader Impact section):**
  The Broader Impact statement has been updated to acknowledge the potential fairness implications of biased compressors under heterogeneous data, and highlights societal benefits and directions for mitigating fairness risks in future research.

All blue-highlighted text in the revised manuscript corresponds to new or substantially revised content. We hope these changes sufficiently address the reviewers’ concerns and improve the clarity and quality of the paper.

---

> ### Author Response · Authors · 2025-07-09
> **Correction of Section References in Summary of Changes**
>
> We would like to issue a clarification and correction regarding the section references in our previous official comment titled “Summary of Changes” (submitted on 09 July 2025).
>
> Due to a reorganization of the manuscript structure, several section indices were misstated:
>
> - “Convergence Rate Analysis (Section 5.6)” should be “Appendix D.4”.
> - “Wall-Clock Computation Overhead (Section 5.3, Appendix D)” should be “Appendix D.5”.
> - “Sign-Based Methods in Related Work (Section 2)” should be “Appendix B.1”.
> - “Broader Impact and Fairness (Broader Impact section)” should be “Appendix A”.
>
> The revised manuscript includes the correct structure and numbering. We apologize for any confusion caused and sincerely thank the reviewers and Action Editor for their understanding.

---

### Decision · Action_Editor_grTU · 2025-08-30

**Recommendation:** Accept as is

**Additional Comments:**

All reviewers lean towards acceptance, since the paper makes valid contributions in proposing an algorithm that combines biased compression with Scaffnew and shows improved communication-accuracy tradeoffs. While it lacks theoretical guarantees to allow for biased compressors in this setting, that can be left for future work.

**Audience:**

Yes

**Audience Explanation:**

This submission should be of interest to the optimization and distributed learning community.

**Claims And Evidence:**

Yes

**Claims Explanation:**

This paper extends the previous Scaffnew work (which shows the benefits of local updates using a prox skip mechanism) by incorporating biased compressors such as top-k and quantization to reduce communication and validates their effectiveness. The author responses addresses several concerns around adding experiments on large-scale settings and improving presentation. The paper claims it demonstrates the empirical convergence of biased compressors, which matches the experiments and algorithm design.